# GAKD: Generative Adversarial Knowledge Distillation For Large Language Models

## Abstract

Current white-box knowledge distillation (KD) methods for large language models (LLMs) often rely on distribution distance metrics, such as forward or reverse Kullback–Leibler Divergence (KLD), as optimization objectives. However, KLD objective only provides token-wise feedback during knowledge distillation, lacking long-range, sequence-level signals and leading to poor distribution alignment between the teacher and student models. To address this, we propose the Generative Adversarial Knowledge Distillation (GAKD) framework, which adopts a minimax adversarial strategy. Specifically, GAKD trains: (1) a generator (student) to align with the teacher's distribution via a combination of sequence-level adversarial loss and reverse KLD loss, and (2) a discriminator to distinguish whether per-token logits are from the teacher or student. By jointly minimizing the token-level reverse KLD and sequence-level adversarial losses, GAKD enables the student model to more effectively align with the teacher's distribution, leading to improved performance. Furthermore, we provide a mathematical proof of the feasibility of optimizing reverse KLD loss on teacher-generated sequences, establishing the theoretical soundness of GAKD. Experimental results on the instruction-following tasks, conducted on the Qwen-3 model families (with parameters ranging from 0.6B to 8B), demonstrate that utilizing the sequence-level signals, GAKD generates more accurate responses than the SOTA baselines, especially in the long-text generation scenario. Our code can be found in `https://anonymous.4open.science/r/GAKD-8753/`.

## 1 Introduction

Auto regressive language models have demonstrated remarkable performance across various natural language processing tasks Vaswani et al. (2017); Brown et al. (2020). However, as LLMs scale up, their computational cost during inference becomes increasingly prohibitive Kaplan et al. (2020). Knowledge distillation (KD) is an effective model compression technique that addresses this challenge by transferring the knowledge of a larger model (teacher) to a smaller model (student) Hinton et al. (2015); Sanh et al. (2019), improving computational efficiency while maintaining competitive performance. The methods of knowledge distillation can generally be categorized into two classes: black-box and white-box approaches. In black-box KD, the teacher model is typically closed-source (e.g., GPT-4 Achiam et al. (2023)), where only its generated text is accessible. In contrast, white-box KD is applied in scenarios where the teacher model is open-source, allowing access to not only its output distribution over generated text but also its intermediate representations Gou et al. (2021).

With the growing trend of open-sourcing large language models Touvron et al. (2023); Bai et al. (2023); Liu et al. (2024), white-box knowledge distillation (KD) has received increasing attention. This approach provides access to richer information from the teacher model, such as its output distribution and hidden representations, enabling more effective distillation. Most existing white-box KD methods adopt Kullback-Leibler (KL) divergence as the optimization objective Kim & Rush (2016): given the teacher distribution $p(y|x)$ and student distribution $q_\theta(y|x)$, these methods primarily aim to minimize either the forward KL divergence $KL(p|q_\theta)$ or the reverse KL divergence $KL(q_\theta|p)$ to align the student model's distribution with that of the teacher. As KL divergence is asymmetric Malinin & Gales (2019) ($KL(p|q_\theta) \neq KL(q_\theta|p)$), prior studies Minka et al. (2005)

have highlighted that forward KL divergence often leads to mode averaging in the context of LLM distillation, where the student distribution $q_\theta$ assigns unreasonably high probabilities to void regions of the teacher distribution $p$. In comparison, reverse KL divergence avoids this issue and has been adopted in certain methods.

Despite its advantages, white-box KD methods face two key limitations: First, these methods provide only **token-wise feedback** during distillation, **lacking signals about long-range consistency or higher-order dependencies across the sequence**. The second challenge stems from the **theoretical requirement of sampling output sequences from the student model** during training when optimizing reverse KL divergence. However, as the student model evolves dynamically during training and generates low-quality sequences in the early stages, this inconsistency negatively impacts the overall distillation performance.

To address the limitations of traditional token-level knowledge distillation, we propose Generative Adversarial Knowledge Distillation (GAKD), a framework that reformulates distillation as a minimax optimization problem inspired by GANs. In GAKD, the student acts as a generator, learning to approximate the teacher's output distribution by jointly optimizing the reverse KLD loss and the **sequence-level** adversarial loss, while a discriminator distinguishes whether logits of a sequence are generated by the teacher or student. The reverse KLD provides local alignment, while the adversarial loss offers global, sequence-level feedback, encouraging the student to holistically align with the teacher's distribution and address issues of long-range consistency. Additionally, we prove that output sequences minimizing reverse KLD can be sampled from the teacher using **importance sampling**, providing a solid theoretical foundation for GAKD's validity.

The contributions of this paper are summarized as follows:

- We propose Generative Adversarial Knowledge Distillation (GAKD), a novel framework that combines reverse KLD with an adversarial objective to provide both token-wise feedback for local alignment and sequence-level feedback for long-range consistency.
- We establish a theoretical proof showing that through importance sampling, the output sequences used to minimize the reverse KLD can be directly sampled from the teacher model, providing a rigorous foundation for the validity and consistency of GAKD.
- Experiments on Qwen-3 models show that GAKD, with sequence-level signals, outperforms SOTA baselines on instruction-following tasks, especially for long-text generation.

## 2 BACKGROUND AND RELATED WORK

### 2.1 KNOWLEDGE DISTILLATION FOR AUTO-REGRESSIVE LM

Knowledge distillation methods for auto-regressive language models can be broadly categorized into two types based on the accessibility of the teacher model: black-box methods and white-box methods Gou et al. (2021). In black-box methods, the teacher models are typically proprietary and closed-source. These methods rely on the output sequences generated by the teacher model, which are then used as the training corpus for the student model Chiang et al. (2023); Shridhar et al. (2023); Hsieh et al. (2023); Kang et al. (2023).

On the other hand, white-box methods leverage internal details of the teacher model. These methods distill either the token-wise output distributions Wen et al. (2023); Jiang et al. (2023); Gu et al. (2023); Agarwal et al. (2024); Ko et al. (2024); Wu et al. (2024); Gu et al. (2024); Boizard et al. (2024) or the intermediate hidden representations Liang et al. (2023); Wang et al. (2024) from the teacher model into the student model. This is achieved through optimization objectives such as forward KL divergence Kim et al. (2023), reverse KL divergence Gu et al. (2023), or their variations Agarwal et al. (2024). Denote the distributions of the teacher and student models as $p(y|x)$ and $q_\theta(y|x)$ respectively, the forward KL divergence is:

$$\mathrm{KL}(p|q_\theta) = p(y|x) \log \frac{p(y|x)}{q_\theta(y|x)}. \tag{1}$$

For a language model, given the input $x$, the corresponding response $\mathrm{y} = \{y_t\}_{t=1}^{T}$ ($T$ is the length of the response) and the token vocabulary $\{Y_1, Y_2, ..., Y_V\}$ ($V$ is the size of the vocabulary), the KLD-

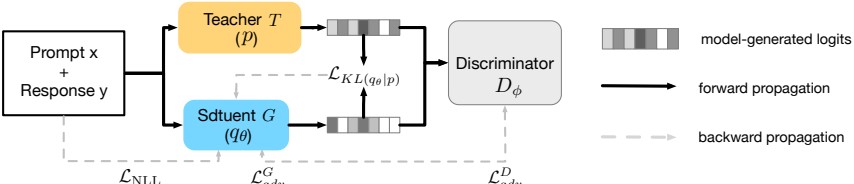

Figure 1: Overview of GAKD.

based distillation of the LM can be decomposed into the summation of the token-wise distillation:

$$\mathcal{L}_{KL(p|q_\theta)} = \sum_{t=1}^{T} \sum_{j=1}^{V} p(Y_j|y_{<t}) \log \frac{p(Y_j|y_{<t})}{q_\theta(Y_j|y_{<t})}. \tag{2}$$

## 2.2 GENERATIVE ADVERSARIA NETWORK

Generative Adversarial Networks (GANs) Goodfellow et al. (2014), have been widely applied across various domains, including image generation, style transfer and data augmentation. The core idea of GANs is to train two neural networks, a generator $G$ and a discriminator $D$, in a minimax game framework. The generator $G$ learns to map a random noise vector $z$ to a data distribution $p_{\text{data}}$, while the discriminator $D$ learns to distinguish between real samples from $p_{\text{data}}$ and fake samples generated by $G$. The two models are trained adversarially: $G$ aims to "fool" $D$, while $D$ improves its ability to classify real versus fake data. The objective function of GANs is formulated as follows:

$$\min_G \max_D \mathbb{E}_{x \sim p_{\text{data}}}[\log D(x)] + \mathbb{E}_{z \sim p_z}[\log(1 - D(G(z)))] \tag{3}$$

where $D(x)$ represents the probability that $x$ is a real sample, and $G(z)$ generates fake samples from the latent noise distribution $p_z$.

GANs have seen significant advancements, addressing challenges like mode collapse, instability, and scalability through diverse innovations Gui et al. (2021). Relativistic GANs (RGANs) Jolicoeur-Martineau (2018) redefine the discriminator to compare the realism of real and fake samples, effectively improving stability and mode coverage. WGANs Arjovsky et al. (2017) optimize Wasserstein distance for smoother gradients, with WGAN-GP Gulrajani et al. (2017) adding gradient penalties for further stability. StyleGAN Karras et al. (2019) revolutionizes image synthesis with hierarchical style control, while BigGAN Brock et al. (2018) pushes scalability to larger datasets. Attention-based models like SAGAN Zhang et al. (2019) improve global context modeling, and CGANs Mirza & Osindero (2014) enable controlled generation with conditional inputs.

## 3 GENERATIVE ADVERSARIAL KNOWLEDGE DISTILLATION

### 3.1 OVERVIEW

Figure 1 illustrates the overall optimization flow of Generative Adversarial Knowledge Distillation (GAKD). In this framework, the teacher model $T$ (a frozen LLM with the probability distribution $p$) and the student model $G$ (a learnable LLM with the probability distribution $q_\theta$) process the same input, which consists of a prompt $x$ concatenated with a response $y$. Both models output logits for the tokens in $x||y$ ($||$ denotes concatenation), which reflect their respective distributions. These logits are then passed to a discriminator $D$, which is trained to distinguish between the teacher's logits and the student's logits. The discriminator provides an adversarial signal $\mathcal{L}_{\text{adv}}^D$ that encourages the student to generate logits that are indistinguishable from those of the teacher at a **sequence level**.

Meanwhile, the student model $q_\theta$ is optimized using three complementary objectives. First, hard-label supervision via the negative log-likelihood $\mathcal{L}_{\text{NNL}}$ of the gold response $y$ ensures the student learns directly from the ground truth. Second, reverse KL divergence $\mathcal{L}_{KL(q_\theta|p)}$ aligns the student's **token-level** predictions with the teacher's output distribution. Third, the adversarial loss $\mathcal{L}_{\text{adv}}^G$ from the discriminator $D_\phi$ encourages **sequence-level** similarity between the student and the teacher. Throughout this process, the teacher model remains fixed, providing a stable target distribution,

while the discriminator and the student are updated iteratively according to their respective objectives. This optimization flow allows the student to progressively align with the teacher's behavior at both token and sequence levels, achieving effective knowledge distillation.

## 3.2 Discriminator Model Design

In Generative Adversarial Knowledge Distillation (GAKD), the role of the discriminator model is to determine whether a given logits sequence originates from the teacher model $T$ or the student model $G$. Specifically, for an input sample consisting of a prompt $x$ and its corresponding response $y$, the input $x||y$ is encoded by either the teacher model ($T$) or the student model ($G$) to produce the logits sequence (denoted as $T(x||y)$ or $G_\theta(x||y)$). The discriminator then classifies the logits as being generated by the teacher or the student. Through this process, the discriminator provides sequence-level, global feedback to the student model via gradients, enabling the student to better approximate the teacher model's distribution.

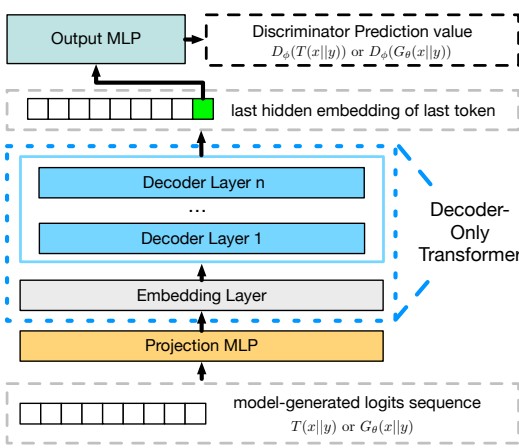

Figure 2: Model architecture of the discriminator.

The discriminator model is designed to handle logits sequences of arbitrary lengths, as the sequence lengths may vary across different training samples. Furthermore, it must possess robust language-level understanding to effectively capture the underlying distributional characteristics of the logits. To meet these requirements, we design our discriminator model architecture based on the **decoder-only transformer**. Figure 2 illustrates the model architecture of the discriminator. Given a logit sequence $T(x||y)$ or $G_\theta(x||y)$ produced by either the teacher model ($T$) or the student model ($G$), a projection MLP is first applied to map the logits into the same dimensionality as required by the decoder-only transformer. The projected logits are then fed into the decoder-only transformer, which consists of two main components: an embedding layer and a decoder block composed of n stacked decoder layers. The decoder-only transformer processes the input and outputs the last hidden embedding of the final token. **This embedding, due to the causal attention mechanism, captures the complete information of the entire logit sequence**. Subsequently, this final embedding is passed through an output MLP layer, which produces the discriminator's prediction—typically a scalar value.

## 3.3 Objective

As is shown in Figure 1, GAKD adopts the generative adversarial training scheme, with the loss $\mathcal{L}_{\text{adv}}^{G}$ updating the generator (student model) $G$ and $\mathcal{L}_{\text{adv}}^{D}$ updating the discriminator $D$. However, directly applying the traditional GAN formulation in equation 3 suffers from issues such as mode collapse and training stability Jolicoeur-Martineau (2018). To address these issues, relativistic GAN (RGAN) Jolicoeur-Martineau (2018) replaces the absolute discriminator with a relativistic one that compares the relative authenticity between real and fake samples rather than judging samples in isolation. The general RGAN objective is defined as:

$$\min_G \max_D V(D, G) = \mathbb{E}_{(p,z) \sim (p_D, p_z)}[f(D_\phi(G_\theta(z)) - D_\phi(p))], \tag{4}$$

where $p_D$ is the real data distribution and $p_z$ is the noise distribution.

This mechanism in RGAN ensures that both real and fake samples participate actively in training dynamics. As the generator improves fake samples $G_\theta(z)$, it simultaneously affects the discriminator's evaluation of real samples $p$ through the relative comparison $D_\phi(G_\theta(z)) - D_\phi(p)$), reducing model collapse and promoting both stability and diversity Huang et al. (2025). The discriminator loss $\mathcal{L}_{D}^{RGAN}$ and generator loss $\mathcal{L}_{G}^{RGAN}$ are defined as follows:

$$\mathcal{L}_{D}^{RGAN} = \mathbb{E}_{(p,z) \sim (p_D, p_z)}[f(D_\phi(p) - D_\phi(G_\theta(z)))], \tag{5}$$

$$\mathcal{L}_{G}^{RGAN} = \mathbb{E}_{(p,z) \sim (p_D, p_z)}[f(D_\phi(G_\theta(z)) - D_\phi(p))], \tag{6}$$

where $f(x) = \log(1 + e^x)$, is the softplus activation function:

In GAKD, we adopt the mechanism of RGAN for training stability and diversity, and the corresponding optimization objective is formulated in Equation 7:

$$\min_\theta \max_\phi \mathbb{E}_{(x,y)\in\mathcal{C}}[f(D_\phi(T(x||y) - D_\phi(G_\theta(x||y)))] + \mathbb{E}_{(x,y)\in\mathcal{C}}[f(D_\phi(G_\theta(x||y)) - D_\phi(T(x||y)))]+$$

$$\mathbb{E}_{x\in\mathcal{C},y\in q_\theta}[\log \frac{q_\theta(y|x)}{p(y|x)}] + \mathbb{E}_{(x,y)\in\mathcal{C}}[-\log q_\theta(y|x)], \tag{7}$$

where $p(y|x)$ and $q_\theta(y|x)$ represent the teacher and student model's output distributions, $T(.)$ and $G_\theta(.)$ denote the output logits by the teacher and student model, $\mathcal{C}$ denotes the training corpus and $||$ stands for concatenation. The first two terms in Equation 7 constitute the RGAN losses for the discriminator $D_\phi$ and generator $G_\theta$, respectively, which together provide sequence-level feedback and align the student's output distribution $q_\theta$ with the teacher's distribution $p$ through a minimax training framework. The third term is the reverse KL divergence, which encourages the student to match the teacher's soft targets (i.e., output distribution) while mitigating issues such as mode averaging. Finally, the last term represents the negative log likelihood (NLL) loss, corresponding to the cross-entropy between the student's predictions and the ground-truth labels (hard targets); incorporating both soft-target and hard-target losses, as advocated in knowledge distillation Hinton et al. (2015), has been shown to improve overall model performance.

Observe that in the first, second, and last expectation terms of Equation 7, both $x$ and $y$ are drawn from the training corpus $\mathcal{C}$[1]. In contrast, for the third term, the expectation corresponding to the reverse KL divergence, $y$ is required to be sampled from the student distribution $q_\theta$:

$$\text{KL}(q_\theta(y|x)\|p(y|x)) = \mathbb{E}_{y\sim q_\theta(\cdot|x)}\left[\log \frac{q_\theta(y|x)}{p(y|x)}\right]. \tag{8}$$

However, sampling from the student model $q_\theta$ for the third term (reverse KL divergence) introduces two drawbacks. First, the other three terms in Equation 7 are sampled from the real distribution (or equivalently, the teacher distribution), while the reverse KL term requires samples from the student model. This discrepancy in sampling sources makes it challenging to perform unified and efficient batch optimization during training. Second, since the student model evolves dynamically throughout training and tends to generate low-quality sequences in the early stages, this inconsistency can adversely affect the overall distillation performance. To overcome the above limitations, we first propose a corollary:

**Corollary 1** *By importance sampling, the reverse KL divergence can be estimated using samples drawn from the teacher model $p(y|x)$.*

**Proof 1** *Let the reverse KL divergence be defined as*

$$\text{KL}(q_\theta(y|x)\|p(y|x)) = \mathbb{E}_{y\sim q_\theta(\cdot|x)}\left[\log \frac{q_\theta(y|x)}{p(y|x)}\right].$$

*For any function $f(y)$, we have*

$$\mathbb{E}_{y\sim q_\theta(\cdot|x)}[f(y)] = \mathbb{E}_{y\sim p(\cdot|x)}\left[\frac{q_\theta(y|x)}{p(y|x)}f(y)\right],$$

*provided that $p(y|x) > 0$ whenever $q_\theta(y|x) > 0$. Applying this to the reverse KL objective, we obtain*

$$\text{KL}(q_\theta(y|x)\|p(y|x)) = \mathbb{E}_{y\sim p(\cdot|x)}\left[\frac{q_\theta(y|x)}{p(y|x)}\log \frac{q_\theta(y|x)}{p(y|x)}\right].$$

*Thus, the reverse KL divergence can be estimated using samples from the teacher model $p(y|x)$ via importance sampling.*

---

[1]Sampling from the training corpus $\mathcal{C}$ can be regarded as sampling from the teacher model if the teacher is sufficiently strong, i.e., the empirical data distribution and the teacher model distribution are approximately the same.

The computational details for the reverse KL divergence derived in Corollary 1 during model training can be found in the Appendix. Utilizing corollary 1 and replacing the third item in equation 7, we obtain the ultimate objective of GAKD in equation 9:

$$\min_\theta \max_\phi \mathop{\mathbb{E}}_{(x,y)\in\mathcal{C}} [f(D_\phi(T(x||y)) - D_\phi(G_\theta(x||y))) + f(D_\phi(G_\theta(x||y)) - D_\phi(T(x||y))) + \\ \frac{q_\theta(y|x)}{p(y|x)} \log \frac{q_\theta(y|x)}{p(y|x)} - \log q_\theta(y|x)]. \tag{9}$$

By proposing and proving corollary 1, we prove that the reverse KLD loss can be estimated and optimized using the training corpus sampled from the teacher output distribution $p$, which overcomes the mode average issue of forward KLD and provides theoretical support for the objective function in equation 9.

### 3.4 OPTIMIZATION ALGORITHM

We propose Algorithm 1 to train the student model to optimize the objective in equation 9 in the generative adversarial manner. The training alternates between updating the discriminator $D_\phi$ and the student model $G_\theta$. For each training iteration, the discriminator is first trained for $k$ steps to distinguish between teacher output logits $T(x||y)$ and student output logits $G_\theta(x||y)$ using a softplus-based objective. Then, the student model is updated using a composite loss that combines three terms: negative log-likelihood loss $\mathcal{L}_{\text{NLL}}$ for task performance, reverse KL divergence loss $\mathcal{L}_{\text{KL}}$ for knowledge distillation, and adversarial loss $\mathcal{L}_{\text{adv}}^q$ to fool the discriminator. The hyperparameters $\alpha$ and $\beta$ control the relative weights of knowledge distillation and adversarial training, respectively.

---

**Algorithm 1** GAKD Training Optimization Algorithm

---

**Require:** Training data $\mathcal{C} = \{(x, y)\}$. Hyperparameters $\alpha$ and $\beta$. Softplus function $f(x)$
    **for** each training iteration **do**
        **for** $k$ steps **do**
            Sample minibatch $[(x_1, y_1), ..., (x_m, y_m)]$ from $\mathcal{C}$
            Update discriminator by ascending its gradient:

$$\nabla_\phi \frac{1}{m} \sum_{i=1}^m f\left(D_\phi(T(x_i||y_i)) - D_\phi(G_\theta(x_i||y_i))\right)$$

        **end for**
        Sample minibatch $[(x_1, y_1), ..., (x_m, y_m)]$ from $\mathcal{C}$
        Compute $\mathcal{L}_{\text{NLL}} = \frac{1}{m} \sum_{i=1}^m - \log q_\theta(y_i|x_i)$
        Compute $\mathcal{L}_{\text{KL}} = \frac{1}{m} \sum_{i=1}^m \frac{q_\theta(y_i|x_i)}{p(y_i|x_i)} \log \frac{q_\theta(y_i|x_i)}{p(y_i|x_i)}$
        Compute $\mathcal{L}_{\text{adv}}^G = \frac{1}{m} \sum_{i=1}^m f\left(D_\phi(G_\theta(x_i||y_i)) - D_\phi(T(x_i||y_i))\right)$
        Update student model by descending its gradient:

$$\nabla_\theta [\frac{1}{2} \cdot \mathcal{L}_{\text{NLL}} + \frac{\alpha}{2} \cdot \mathcal{L}_{\text{KL}} + \beta \cdot \mathcal{L}_{\text{adv}}^G]$$

    **end for**

---

## 4 EXPERIMENTS

### 4.1 EXPERIMENTAL SETUP

**Base Models** We adopt the Qwen3 Yang et al. (2025a) architecture according to its demonstrated superiority over similar-sized baselines including Llama-3 Grattafiori et al. (2024), Gemma-3 Kamath et al. (2025), and Qwen2.5 Yang et al. (2025b) variants across comprehensive benchmarks. We use Qwen3-8B as the teacher model and distill its knowledge to Qwen3 models with 0.6B, 1.7B and 4B parameters.

| #Params | Method | DollyEval | | | SelfInst | | | S-NI | | | UnNI | | |
|---|---|---|---|---|---|---|---|---|---|---|---|---|---|
| | | Rouge-L | BLEU | Exact Match | Rouge-L | BLEU | Exact Match | Rouge-L | BLEU | Exact Match | Rouge-L | BLEU | Exact Match |
| 8B | teacher | 32.826 | 11.408 | 5.200 | 28.595 | 8.218 | 6.612 | 46.480 | 18.237 | 0.826 | 40.712 | 17.850 | 3.663 |
| 0.6B | SFT | 28.477 | **10.376** | **3.400** | 19.260 | 5.205 | **2.479** | 31.301 | 9.150 | 0.000 | 35.367 | 13.314 | 1.372 |
| | FKLD | 28.632 | 9.821 | 2.600 | 21.329 | **6.517** | 2.066 | 34.458 | 10.930 | **0.118** | 36.931 | 14.310 | 1.539 |
| | RKLD | 28.024 | 9.934 | 2.800 | 22.099 | 6.189 | 1.653 | **35.990** | 11.334 | **0.118** | **38.973** | 15.598 | 1.923 |
| | MiniLLM | 26.807 | 8.566 | 2.600 | 19.238 | 5.011 | 1.240 | 33.335 | 10.313 | **0.118** | 35.453 | 13.140 | 1.363 |
| | GAKD w. FKL | 27.566 | 8.976 | 2.800 | 20.001 | 5.363 | 0.413 | 34.249 | 10.510 | **0.118** | 37.243 | 14.192 | 1.376 |
| | GAKD (ours) | **28.747** | 10.221 | **3.400** | **22.780** | 6.337 | 1.653 | 35.959 | **11.541** | **0.118** | 38.948 | **15.768** | **1.974** |
| | △ | 0.115 | -0.155 | 0.000 | 0.681 | -0.180 | -0.826 | -0.031 | 0.207 | 0.000 | -0.025 | 0.170 | 0.051 |
| 1.7B | SFT | 30.395 | 10.858 | 4.400 | 26.543 | 7.952 | 4.546 | 43.108 | 15.765 | 0.472 | 40.200 | 17.055 | 3.136 |
| | FKLD | 31.350 | 11.092 | 4.400 | 25.365 | 7.123 | **4.959** | 42.741 | 15.496 | 0.413 | 40.484 | 17.369 | 3.065 |
| | RKLD | 30.129 | 10.548 | 3.600 | 27.167 | 7.633 | 4.546 | 43.894 | 15.918 | 0.531 | 42.111* | 18.591* | 3.563 |
| | MiniLLM | 28.680 | 9.477 | 4.200 | 25.202 | 7.399 | 4.132 | 43.091 | 15.572 | **0.708** | 40.610 | 17.475 | 3.103 |
| | GAKD w. FKL | **31.385** | 11.145 | 4.200 | 24.598 | 6.707 | 4.546 | 43.932 | 15.965 | 0.354 | 40.807* | 17.704 | 3.316 |
| | GAKD (ours) | 31.231 | **11.337** | **4.600** | **27.342** | **8.004** | **4.959** | **44.440** | **15.990** | **0.708** | **42.918*** | **19.159*** | **3.638** |
| | △ | -0.154 | 0.192 | 0.200 | 0.175 | 0.052 | 0.000 | 0.508 | 0.025 | 0.000 | 0.807 | 0.568 | 0.075 |
| 4B | SFT | 31.468 | 10.847 | 4.200 | 28.373 | 7.554 | 5.372 | 44.801 | 17.051 | 0.708 | 40.495 | 17.465 | 3.575 |
| | FKLD | 31.858 | 10.779 | 4.600 | **28.970*** | 7.639 | **7.025*** | 46.340 | 18.065 | 0.531 | 40.731* | 17.588 | 3.730* |
| | RKLD | 31.747 | 11.001 | 5.000 | 27.963 | 7.657 | 5.372 | 46.797* | 18.370* | 0.826 | 41.634* | 18.311* | 3.905* |
| | MiniLLM | 31.859 | **11.065** | **5.600*** | 27.332 | 7.409 | 4.959 | 45.950 | 17.927 | 0.649 | 40.741* | 17.590 | 3.604 |
| | GAKD w. FKL | 30.529 | 9.923 | 4.400 | 27.557 | 7.684 | 5.785 | 45.713 | 17.297 | 0.708 | 41.266* | 17.955* | 3.625 |
| | GAKD (ours) | **32.368** | 10.794 | 5.400 | 28.389 | **7.976** | 5.372 | **47.206*** | **18.737*** | **1.122*** | **41.969*** | **18.554*** | **4.077*** |
| | △ | 0.509 | -0.271 | -0.200 | -0.581 | 0.292 | -1.653 | 0.409 | 0.367 | 0.296 | 0.335 | 0.243 | 0.127 |

Table 1: Evaluation results on Qwen-3 model series. △ denotes the performance improvement of GAKD relative to the baselines. The best scores of each model size are **boldfaced**, and the scores where the student model outperforms the teacher are marked with *.

**Training** Experiments are conducted using the Dolly[2] dataset, comprising 12.5K human-annotated instruction-response pairs allocated for training, with 0.5K and 1K samples designated for testing and validation phases, respectively. The dataset encompasses seven distinct task categories: Creative Writing, Closed QA, Open QA, Summarization, Information Extraction, Classification, and Brainstorming, thereby ensuring comprehensive coverage of instruction-following paradigms and task diversity. More training details can be found in the Appendix.

**Baselines** We compare GAKD against 4 baselines: 1) **SFT w/o KD** fine-tunes the student model supervised by the golden responses, 2) **FKLD** fine-tunes the student model by minimizing the forward KL divergence between $p$ and $q_\theta$ at token-level. 3) **RKLD** fine-tunes the student model by minimizing the reverse KL divergence between $p$ and $q_\theta$ at token-level and 4) **MiniLLM** Gu et al. (2023) minimizes the reverse KL divergence between $p$ and $q$ through a customized, PPO-based approach.

**Evaluation Datasets** Following the benchmark construction methodology in MiniLLM Gu et al. (2023), we evaluate our distilled models on four instruction-following benchmarks: 1) **DollyEval** contains 500 questions sampled from human-annotated dataset Dolly, 2) **SelfInst**[3] is a user-oriented evaluation set with 252 diverse question samples, 3) **S-NI**[4] is constructed from the test set of Super-Natural-Instructions Wang et al. (2022). We split this set into 3 subsets whose ground truth response lengths lies in $[0, 5]$, $[6, 10]$ and $[11, +\infty]$, and use the $[11, +\infty]$ set for long-text generation evaluation in the following experiments. 4) **UnNI**[5]: constructed from Unnatural-Instructions Honovich et al. (2022). Similar to S-NI, we use the $[11, +\infty]$ set for long-text generation evaluation in the following experiments.

**Evaluation Metrics** We use three metrics for comprehensive assessment: 1) **Rouge-L** Lin (2004) measures the longest common subsequence between generated and reference text, capturing content overlap, 2) **Exact Match**: evaluates precise answer correspondence, requiring complete string matching, and 3) **BLEU** Papineni et al. (2002): assesses n-gram overlap quality between generated and reference responses.

## 4.2 EXPERIMENTAL RESULTS

We evaluate and compare GAKD with all baselines on the 4 evaluation datasets. For comparative analysis, we replace the reverse KLD loss $\mathcal{L}_{KL(q_\theta|p)}$ in GAKD by the forward KLD loss $\mathcal{L}_{KL(p|q_\theta)}$ and denote this implementation as GAKD w. FKL. The evaluation results are presented in Table 1.

---

[2]https://hf-mirror.com/datasets/MiniLLM/dolly

[3]https://hf-mirror.com/datasets/MiniLLM/self-inst

[4]https://hf-mirror.com/datasets/MiniLLM/sinst

[5]https://hf-mirror.com/datasets/MiniLLM/uinst

| | | DollyEval | | | SelfInst | | | S-NI | | | UnNI | | |
|---|---|---|---|---|---|---|---|---|---|---|---|---|---|
| | | Rouge-L | BLEU | Exact Match | Rouge-L | BLEU | Exact Match | Rouge-L | BLEU | Exact Match | Rouge-L | BLEU | Exact Match |
| $\alpha$ | 15 | 31.082 | 11.129 | 4.200 | 26.942 | **8.027** | 4.132 | 43.766 | 15.730 | 0.472 | 42.235 | 18.849 | 3.508 |
| | 10 | **31.231** | **11.337** | 4.600 | **27.342** | 8.004 | **4.959** | 44.440 | **15.990** | **0.708** | **42.918** | 19.159 | 3.638 |
| | 5 | 31.038 | 10.704 | **5.000** | 26.247 | 7.340 | 3.719 | 43.363 | 15.490 | **0.708** | 42.262 | 18.898 | 3.466 |
| | 1 | 30.034 | 10.004 | 4.000 | 24.749 | 7.373 | 3.719 | 43.618 | 15.575 | 0.413 | 42.867 | 19.248 | 3.529 |
| $\beta$ | 0.9 | 30.080 | 10.846 | 4.000 | 24.183 | 7.213 | 2.479 | 43.397 | 15.076 | 0.413 | 42.698 | 19.245 | 4.119 |
| | 0.5 | 30.633 | 10.396 | 4.000 | 26.665 | 7.987 | 4.959 | **44.452** | **16.302** | 0.649 | 42.867 | **19.248** | 3.529 |
| | 0.1 | 31.231 | **11.337** | 4.600 | 27.342 | 8.004 | 4.959 | 44.440 | 15.990 | 0.708 | 42.918 | 19.159 | 3.638 |
| | 0.05 | 30.848 | 10.806 | 4.400 | 27.126 | 7.780 | 4.546 | 43.477 | 15.531 | 0.590 | 42.425 | 19.031 | **3.671** |
| | 0.01 | **31.634** | 11.088 | **4.800** | **27.507** | 7.799 | 4.546 | 44.300 | 15.978 | 0.472 | 42.685 | 19.143 | 3.575 |

Table 2: Performance comparison of GAKD under different hyperparameter settings, using Qwen3-1.7B as the student model. $\alpha$ and $\beta$ are hyperparameters controlling the strengths of the reverse KLD loss ($\mathcal{L}_{KL}$) and generator adversarial loss ($\mathcal{L}_{\text{adv}}^{G}$) in Algorithm 1.

**Consistent Improvements Over Baselines.** As is shown in table 1, GAKD demonstrates remarkable stability across all model sizes (0.6B, 1.7B, and 4B) and datasets, achieving the best performance in most situations. For 0.6B models, GAKD achieves highest Rouge-L scores on DollyEval and SelfInst, and best BLEU and exact match results on S-NI and UnNI. For 1.7B models, it leads in all datasets, notably outperforming the teacher on UnNI. For 4B models, GAKD excels on S-NI and UnNI, surpassing the teacher in multiple metrics. These results highlight GAKD's robustness and scalability across model sizes. Moreover, using reverse KLD rather than forward KLD allows GAKD to achieve consistently better performance than its FKL variant (GAKD w. FKL) across all student model sizes and evaluation sets, highlighting the effectiveness of reverse KLD in our framework. Finally, both GAKD and other baselines exhibit generally low exact match scores on S-NI and UnNI. This is primarily because all questions in S-NI are open-ended, allowing for diverse valid answers, while most questions in UnNI are also open-ended with only a minority being closed-form with fixed answers. As a result, exact match, which requires a strict correspondence to the reference, tends to be low on these datasets. In contrast, metrics that capture semantic similarity, such as Rouge-L and BLEU, remain high, reflecting the models' ability to generate responses that are semantically and structurally aligned with the reference answers.

**Long-Text Generation Excellence.** Remarkably, GAKD achieves the best performance on long-text generation benchmarks such as S-NI and UnNI, where open-ended questions typically require extended and coherent responses. On both the 1.7B and 4B student models, GAKD even surpasses the teacher model on these evaluation sets. This superior performance can be attributed to GAKD's carefully designed adversarial loss, which provides global, sequence-level signals for distillation. These signals effectively guide the student model to capture long-range dependencies and global structure—capabilities that are essential for generating coherent, lengthy, and complex outputs.

### 4.3 ABLATION STUDY

To evaluate the impact of various hyperparameters and GAN variants in GAKD, we conducted a comprehensive ablation study using the Qwen3-1.7B model. The detailed experimental details can be found in the Appendix.

#### 4.3.1 IMPACT OF HYPERPARAMETER SETTINGS

Table 2 details the performance of GAKD under varying weights for the reverse KL loss ($\alpha$), which controls the strength of the token-level signal. Increasing $\alpha$ from 1 to 5 significantly boosts exact match scores on DollyEval and SelfInst, indicating that stronger token-level feedback helps the student model capture local generation patterns. The model achieves its best overall performance as $\alpha$ is increased to 10, where an optimal balance between token-level and sequence-level guidance is struck. However, excessive $\alpha$ values beyond this point disrupt this equilibrium, leading to performance degradation as the model becomes overly constrained by token-level feedback.

The parameter $\beta$ controls the strength of the adversarial loss for the generator (student model). To examine its effect, we conduct ablation studies with different $\beta$ values and plot the training loss curves for both the generator and discriminator in Figure 3. When $\beta$ is set too high (e.g., 0.9), both losses converge rapidly after only brief initial oscillations, indicating an insufficient adversarial process and resulting in poor model performance (as shown in Table 2). Meanwhile if $\beta$ is too low (e.g., 0.01), the adversarial signal becomes too weak, causing persistent oscillations in the losses without

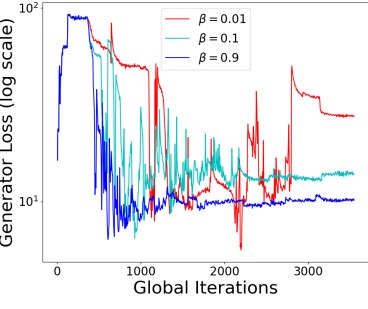
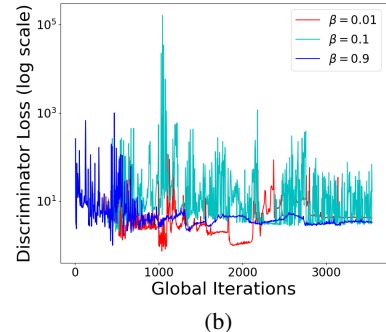

(a)                                              (b)

Figure 3: Training loss curves for (a) generator and (b) discriminator of GAKD, with varying $\beta$ values and Qwen3-1.7B as the student model.

| | DollyEval | | | SelfInst | | | S-NI | | | UnNI | | |
|---|---|---|---|---|---|---|---|---|---|---|---|---|
| GAN Variant | Rouge-L | BLEU | Exact Match | Rouge-L | BLEU | Exact Match | Rouge-L | BLEU | Exact Match | Rouge-L | BLEU | Exact Match |
| GAKD w. GAN | 30.760 | 11.127 | **4.600** | 26.944 | 7.819 | 4.132 | 44.107 | **16.217** | 0.472 | **43.094** | **19.391** | **3.864** |
| GAKD w. WGAN | **31.234** | 11.210 | **4.600** | 27.319 | 7.897 | 4.546 | 43.691 | 15.866 | 0.590 | 41.680 | 18.473 | 3.537 |
| GAKD w. RGAN (ours) | 31.231 | **11.337** | **4.600** | **27.342** | **8.004** | **4.959** | **44.440** | 15.990 | **0.708** | 42.918 | 19.159 | 3.638 |

Table 3: Performance comparison on different GAN variants using Qwen-1.7B as the student model.

convergence, leading to subpar results. Setting $\beta$ to a moderate value (e.g., 0.1) strikes a better balance: the adversarial process is maintained throughout training, with sustained but eventually converging oscillations in the loss curves, ultimately yielding optimal performance.

### 4.3.2 IMPACT OF GAN VARIANTS

To investigate the impact of different generative adversarial strategies on GAKD training, we experiment with three GAN variants within GAKD: standard GAN Goodfellow et al. (2014), WGAN Gulrajani et al. (2017), and RGAN (original). The detailed objective formulations of GAKD w. GAN and GAKD w. WGAN can be found in the Appendix. As shown in Table 3, GAKD with GAN achieves the best performance on the UnNI dataset, while GAKD with WGAN performs comparably to RGAN on DollyEval. Notably, GAKD with RGAN demonstrates more consistent and superior results across all three datasets, outperforming both GAN and WGAN. This advantage can be attributed to RGAN's relative discrimination mechanism, which enhances training stability and focuses on relative rather than absolute judgments.

## 5 CONCLUSION

In this paper, we introduce the Generative Adversarial Knowledge Distillation (GAKD) framework, which fundamentally advances the knowledge distillation paradigm for large language models by integrating an adversarial learning mechanism. By jointly optimizing the reverse KLD loss and a sequence-level adversarial objective, GAKD enables the student model to receive both fine-grained token-level and holistic sequence-level feedback. The adversarial training mechanism, wherein the student and discriminator are engaged in a minimax game, not only encourages the student to closely match the teacher's output distribution, but also promotes better long-range consistency in generation. Our theoretical analysis establishes the validity of optimizing reverse KLD on teacher-generated sequences, and extensive experiments on Qwen-3 model families demonstrate the effectiveness of GAKD, with notable improvements over state-of-the-art baselines, particularly for long-text generation.

### DECLARATION OF LLM USAGE

The usage of LLMs is strictly limited to aid and polish the paper writing

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

# A APPENDIX

## A.1 REVERSE KL LOSS COMPUTATION

We restate the reverse KL divergence of Corollary 1 in a form amenable to mini-batch estimation.

$$\mathrm{KL}\big(q_\theta(y \mid x) \,\|\, p(y \mid x)\big) = \mathbb{E}_{y \sim p(\cdot \mid x)} \left[ \frac{q_\theta(y \mid x)}{p(y \mid x)} \log \frac{q_\theta(y \mid x)}{p(y \mid x)} \right]$$
$$\approx \frac{1}{|\mathcal{C}|} \sum_{(x,y) \in \mathcal{C}} w(x,y)\, \ell(x,y), \tag{10}$$

where the sequence-level importance weight $w$ and the corresponding token-level log-ratio sum $\ell$ are

$$w(x,y) = \prod_{t=1}^{T} \frac{q_\theta\big(y_t \mid x, y_{<t}\big)}{p\big(y_t \mid x, y_{<t}\big)},$$
$$\ell(x,y) = \sum_{t=1}^{T} \log \frac{q_\theta\big(y_t \mid x, y_{<t}\big)}{p\big(y_t \mid x, y_{<t}\big)}. \tag{11}$$

All conditional probabilities needed for $w(x,y)$ and $\ell(x,y)$ are produced by a single forward pass through each model (teacher $p$ and student $q_\theta$) on the full target sequence, so no additional autoregressive sampling loop is required.

## A.2 TRAINING DETAILS

Our experiments are conducted using NVIDIA A100 80GB GPUs, and we employ the Dolly dataset for both training and validation. All experiments use a fixed global batch size of 32 and a maximum length of 512. The implementation is based on PyTorch 2.2.2 with CUDA 12.1, and we utilize the AdamW optimizer for all generator training procedures.

### BASELINES

We conduct training using: SFT w/o KD, FKLD, RKLD, MiniLLM, and our proposed approach. For models with fewer than 1.3B parameters, we tune learning rates across [5e-4, 1e-4, 5e-5], training for 20 epochs. For models with 1.3B or more parameters, we explore learning rates in [5e-5, 1e-5, 5e-6], training for 10 epochs. Optimal hyperparameters are chosen based on the performance of the validation set.

Additionally, training MiniLLM consists of two phases:

- Phase 1: We fine-tune the model for 3 epochs using the best learning rate of the corresponding SFT w/o KD baselines, and select the checkpoint with the lowest validation loss.
- Phase 2: We continuously train the model from Phase 1 using a learning rate $5e-6$ in all cases. We collect 256 sentences at once and adopt 4 inner epochs when doing the policy optimization. We use temperature = 1 when sampling from $q_\theta$. We train the model for 5000 steps and select the final checkpoint using the Rouge-L score on the validation set.

### GAKD TRAINING

We implement a two-stage training process to optimize the model's performance.

**Stage 1:** We fine-tune the model on the Dolly dataset for 3 epochs, adopting the optimal learning rate from the corresponding SFT w/o KD baseline. The checkpoint yielding the lowest validation loss is selected as the foundation for Stage 2. In this stage, training on a single NVIDIA A100 80GB GPU takes approximately 3 hours for Qwen3-0.6B, 2.5 hours for Qwen3-1.7B, and 5 hours for Qwen3-4B.

**Stage 2:** Building on the Stage 1 checkpoint, we employ our GAKD strategy. For models with 1.3B or more parameters, we use learning rates [5e-5, 1e-5, 5e-6] and train for 10 epochs; for models with

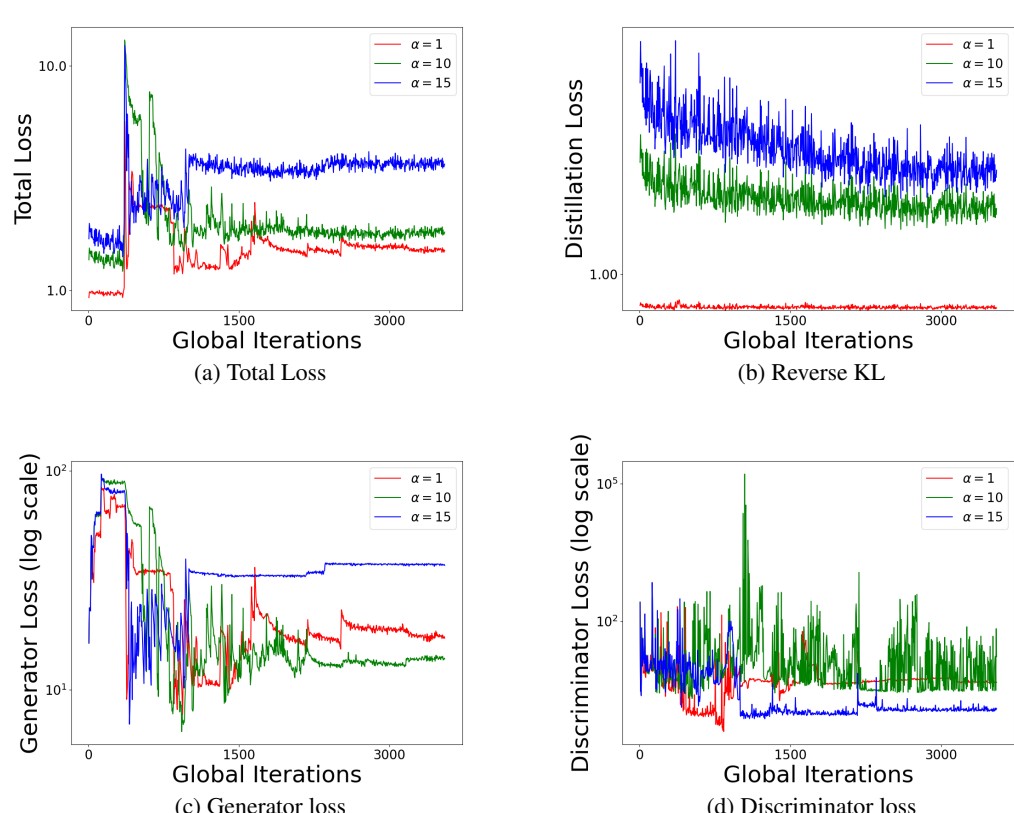

Figure 4: Training Loss (total, reverse KL, generator, discriminator) curves for varying $\alpha$.

fewer than 1.3B parameters, we use learning rates [5e-4, 1e-4, 5e-5] and train for 20 epochs. In the adversarial process, the discriminator is pre-trained on teacher outputs for one epoch, after which the generator and discriminator are updated alternately. The discriminator, selected as Qwen3-0.6B, uses AdamW with a fixed learning rate of 0.0002. The final model is selected based on the highest Rouge-L score on the validation set. Using four NVIDIA A100 80GB GPUs, the training runtime is around 10 hours, 5 hours, 10 hours for Qwen3-0.6B, Qwen3-1.7B and Qwen3-4B, respectively.

### A.3 ABLATION STUDY DETAILS

#### HYPERPARAMETER ABLATION

We conduct ablation studies on the hyperparameters $\alpha$ and $\beta$, with results presented in Table 2. All experiments utilize the Qwen3-0.6B architecture as the discriminator, employing a fixed learning rate of 5e-5, a batch size of 32, training for 10 epochs, and the AdamW optimizer for both the generator and discriminator. During the ablation experiments, when varying $\alpha$, $\beta$ was fixed at 0.1. For the analysis of $\beta$, $\alpha$ were held constant at 10.

The value of $\alpha$ influences training stability and balance. Figure 4 illustrates the effects of varying $\alpha$. When $\alpha$ is too low ($\alpha = 1$), the reverse KL divergence loss $L_{KL}$ contributes weakly to the overall objective, resulting in ineffective minimization of the $L_{KL}$. As $\alpha$ increases, the influence of the $L_{KL}$ becomes more pronounced, evidenced by a declining curve in $L_{KL}$. While this also brings slower convergence of the adversarial components and increased oscillations in the adversarial losses. Conversely, when $\alpha$ is excessively high ($\alpha = 15$), optimization becomes overly dominated by the $L_{KL}$ objective, overshadowing the adversarial loss. This imbalance causes the generator loss to rise and fail to converge, ultimately disrupting the overall training process.

| Critic | DollyEval | | | SelfInst | | | S-NI | | | UnNI | | |
|--------|-----------|------|-------------|----------|------|-------------|------|------|-------------|------|------|-------------|
| | Rouge-L | BLEU | Exact Match | Rouge-L | BLEU | Exact Match | Rouge-L | BLEU | Exact Match | Rouge-L | BLEU | Exact Match |
| Qwen3-0.6B | **31.231** | **11.337** | **4.600** | **27.342** | **8.004** | **4.959** | **44.440** | **15.990** | **0.708** | 42.918 | 19.159 | 3.638 |
| Qwen3-1.7B | 30.745 | 10.635 | 4.200 | 26.692 | 7.990 | 4.132 | 44.004 | 15.931 | **0.708** | **43.092** | **19.458** | **3.721** |

Table 4: Performance comparison for choosing different model sizes for GAKD discriminator, using Qwen3-1.7B as the student model.

## IMPACT OF DISCRIMINATOR MODEL SIZE

To assess the influence of discriminator architecture and size in GAKD, we compare Qwen3-0.6B and Qwen3-1.7B as discriminators, with experimental results summarized in Table 4. Our results show that larger discriminators like Qwen3-1.7B are more effective for long-text generation tasks, as their stronger language understanding allows them to capture complex dependencies and provide richer sequence-level feedback. Conversely, for closed question evaluation, where responses are short and fixed in form, smaller models such as Qwen3-0.6B perform better. Overall, larger discriminator models are preferable for complex, long-text tasks, while smaller ones are better suited to simpler scenarios.

## DISCRIMINATOR AND GAN VARIANTS ABLATION

In the ablation study of discriminator and GAN variants, we fixed the hyperparameters $\alpha$ and $\beta$ to 10 and 0.1, respectively. In GAN Variants ablation, we employed the Qwen3-0.6B model as the discriminator, and learning rate was set to 0.0002 uniformly, with AdamW used as the optimizer for GAN and RGAN, while RMSprop was applied for WGAN to ensure training stability.

## A.4 OBJECTIVE DETAILS FOR GAN VARIANTS ABLATION STUDY

For GAKD with GAN, the discriminator and generator loss functions are:

$$\mathcal{L}_D^{\text{GAN}} = \mathop{\mathbb{E}}_{(x,y)\in\mathcal{C}} \left[\log D_\phi(T(x||y))\right] + \mathop{\mathbb{E}}_{(x,y)\in\mathcal{C}} \left[\log\left(1 - D_\phi(G_\theta(x||y))\right)\right], \tag{12}$$

$$\mathcal{L}_G^{\text{GAN}} = \mathop{\mathbb{E}}_{(x,y)\in\mathcal{C}} \left[\log\left(1 - D_\phi(G_\theta(x||y))\right)\right]. \tag{13}$$

In GAKD with GAN, the discriminator aims to maximize $\mathcal{L}_D^{\text{GAN}}$, while the generator aims to minimize $\mathcal{L}_G^{\text{GAN}}$.

For GAKD with WGAN, the objective functions can be formulated as:

$$\mathcal{L}_D^{\text{WGAN}} = \mathop{\mathbb{E}}_{(x,y)\in\mathcal{C}} \left[D_\phi(T(x||y))\right] - \mathop{\mathbb{E}}_{(x,y)\in\mathcal{C}} \left[D_\phi(G_\theta(x||y))\right], \tag{14}$$

$$\mathcal{L}_G^{\text{WGAN}} = - \mathop{\mathbb{E}}_{(x,y)\in\mathcal{C}} \left[D_\phi(G_\theta(x||y))\right]. \tag{15}$$

where the discriminator $D_\phi$ is constrained to be 1-Lipschitz. The discriminator aims to maximize $\mathcal{L}_D^{\text{WGAN}}$, while the generator aims to minimize $\mathcal{L}_G^{\text{WGAN}}$.

## A.5 VISUALIZATION OF THE TRAINING ADVERSARIAL LOSS FOR GAKD

To better understand the adversarial training behavior of our proposed model, we visualize the training losses of the generator and discriminator across global iterations, as shown in Figure 5. The generator and discriminator losses exhibit distinct dynamics throughout training. During the first epoch, spanning 0 to 352 global iterations, only the discriminator was updated while the generator remained fixed. As a result, the generator loss stayed at a high level, while the discriminator loss showed a decreasing trend. After the first epoch, both the generator and discriminator were jointly trained, leading to a slight increase in discriminator loss and a decrease in generator loss, implying improved generation quality. Although some fluctuations remain, the overall trend suggests that the adversarial training begins to converge.

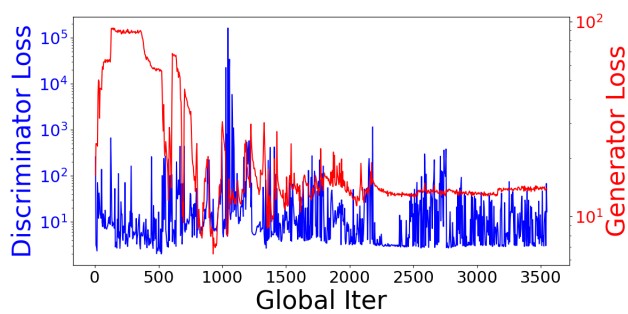

Figure 5: Adversarial loss over global iterations.

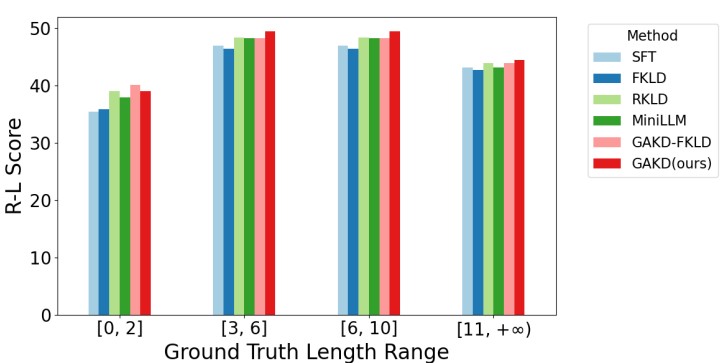

Figure 6: Rouge-L score comparison of GAKD and baselines across different ground truth answer length ranges on S-NI.

### A.6 IMPACT OF GROUND TRUTH RESPONSE LENGTH ON DISTILLATION PERFORMANCE

We evaluate the performance of our proposed GAKD method and various baselines using Qwen3-4B as the student model on the whole S-NI evaluation set (with ground truth response length from 0 to $+\infty$), and stratify the experimental results by the ground truth answer length. As shown in Figure 6, when the ground truth answers are very short (length range $[0, 2]$), the RKLD baseline outperforms GAKD. However, as the ground truth length increases, GAKD consistently demonstrates superior performance across all baselines. These results suggest that GAKD is particularly effective for long-text generation tasks, highlighting its ability to provide more informative sequence-level feedback during the knowledge distillation process.

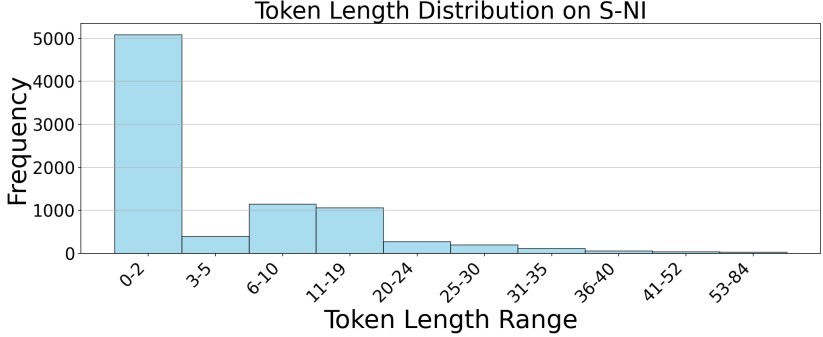

Figure 7: Ground truth response token length distribution of the S-NI dataset.

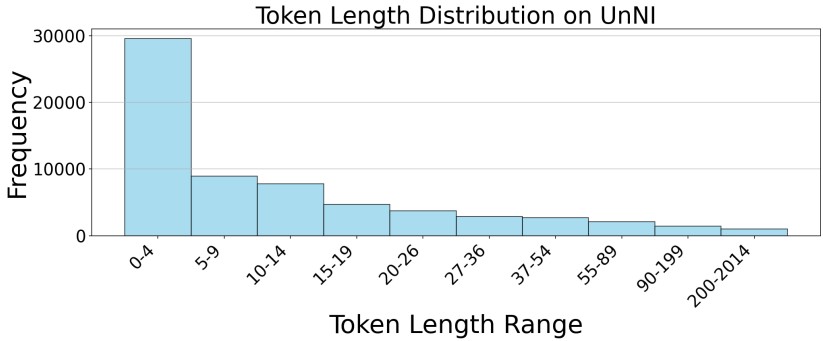

Figure 8: Ground truth response token length distribution of the UnNI dataset.

### A.7 GROUND TRUTH RESPONSE LENGTH DISTRIBUTION ON S-NI AND UNNI

Figure 7 and 8 presents the distributions of ground truth response length for questions in S-NI and UnNI datasets. The S-NI dataset comprises 1694 entries, with a minimum token count of 1 and a maximum of 84. Moreover, the UnNI dataset includes 23,916 entries, spanning from a minimum of 1 to a maximum of 2014 tokens.

| | *Case #1* |
|---|---|
| **Instruction** | Prepare an email signature template for the given person or company. |
| **Input** | a graduate student of computer science |
| **Ground-truth** | [Your name]\n[Your degree] in Computer Science\n[Your university]\n\nWebsite: [Your website]\nORCID ID: [Your ORCID ID] |
| **Teacher** | Hi, I am a computer science graduate student. I am interested in learning more about your company and its offerings. Please let me know if there is any opportunity to work with your team. |
| **SFT** | Name: [insert name]\nEmail: [insert email address]\nPhone: [insert phone number]\nEducation: [insert degree, university, year]\nMajor: [insert major]\nLinkedIn: [insert LinkedIn profile URL]\nGitHub: [insert GitHub profile URL]\nTwitter: [insert Twitter profile URL] |
| **FKLD** | Hi, I'm [Name], a graduate student at [School]. My research is about [research topic]. You can contact me at [email address] or [phone number]. |
| **RKLD** | Hi, I am [Name], a graduate student in computer science. I am interested in machine learning and data science. I have a lot of experience in Python programming and am proficient in machine learning algorithms such as linear regression, logistic regression, decision trees, and random forests. I am also familiar with deep learning frameworks such as TensorFlow and PyTorch. I am always looking for new ways to improve my skills and stay up-to-date with the latest developments in the field. |
| **MiniLLM** | Dear colleagues, friends, and family, \n\nI am an enthusiastic graduate student in computer science at the University of Technology. My research interests include natural language processing, machine learning, and artificial intelligence. I am passionate about solving complex problems and creating innovative solutions that can make a positive impact on society. I am always eager to learn and grow, both professionally and personally. Thank you for your time and consideration. |
| **GAKD w. FKLD** | Dear [name], \n\nI am a graduate student of computer science. My name is [name], and I am a student at [university]. I am currently studying [field of study], and I am interested in [research interests]. I am looking for [type of position], and I am hoping to find a position that aligns with my interests and skills. \n\nIf you are interested in discussing this further, I would be happy to speak with you about [topic]. I can be reached at [email address] or [phone number]. \n\nThank you for your time and consideration. |
| **GAKD (ours)** | Name: [Name]\nTitle: Graduate Student\nInstitution: [Institution]\nEmail: [Email Address]\nPhone: [Phone Number]\nLocation: [Location]\nLinkedIn: [LinkedIn Profile]\nTwitter: [Twitter Profile]\nGitHub: [GitHub Profile]\nWebsite: [Website]\nBio: [Bio] |

Table 5: Case study #1 on instruction-tuning setting, with Qwen3-4B as student model.

*Case #2*

| | |
|---|---|
| **Instruction** | You should choose a YouTube video title based on the video's content. A video's title tells viewers what to expect from it. It should be direct, honest, and clear. The title of the video needs to capture the attention of viewers, so do not use an unclear or ambiguous one. |
| **Input** | A research study has been conducted to determine if exercise really can boostÿour metabolism. |
| **Ground-truth** | You Will Not Believe The Results Of A Recent Study On Exercise And The Metabolism! |
| **Teacher** | Does Exercise Boost Metabolism? Research Study Reveals Surprising Truth |
| **SFT** | "Does exercise boost metabolism?" |
| **FKLD** | Do Exercise to Boost Your Metabolism? A Research Study |
| **RKLD** | This video will tell you the answer to this question. |
| **MiniLLM** | "Does exercise really boost your metabolism? A study reveals the truth!" |
| **GAKD w. FKLD** | Here are some options for a YouTube video title: \n\n1. Does exercise really boost your metabolism? \n2. Does exercise boost your metabolism? \n3. Does exercise boost your metabolic rate? \n4. Can exercise boost your metabolism? |
| **GAKD (ours)** | "Does exercise really boost your metabolism? A study reveals the truth!" |

Table 6: Case study #2 on instruction-tuning setting, with Qwen3-4B as student model.

*Case #3*

| | |
|---|---|
| **Instruction** | You will be given a series of questions and answers. For each answer, you must choose from a list of acceptable synonyms the one that best matches the context of the question. Only select one word for each answer. |
| **Input** | Questions: ['What is Nyx?', 'What is Erebus?']\nAnswers: [['goddess', 'female deity', 'Nike'], ['darkness', 'gloominess', 'place where dead souls go']]. |
| **Ground-truth** | Nyx is a goddess.\nErebus is darkness. |
| **Teacher** | Nyx is a goddess, Erebus is darkness. |
| **SFT** | Nyx is a goddess and a female deity. Erebus is darkness, gloominess, and a place where dead souls go. |
| **FKLD** | Nyx is a goddess, a female deity, and Nike is a synonym for Nyx. Erebus is darkness, gloominess, and a place where dead souls go. |
| **RKLD** | Nyx is a goddess and female deity. Erebus is darkness and gloominess. |
| **MiniLLM** | Nyx and Erebus are both deities. Nyx is a goddess, and Erebus is darkness. |
| **GAKD w. FKLD** | Nyx is a goddess and female deity. Erebus is darkness, gloominess, and a place where dead souls go. |
| **GAKD (ours)** | Nyx is a goddess. Erebus is darkness. |

Table 7: Case study #3 on instruction-tuning setting, with Qwen3-4B as student model.

