# OpenReview forum: "GAKD: Generative Adversarial Knowledge Distillation For Large Language Models"
_ICLR.cc/2026/Conference — Submitted to ICLR 2026_

### Official Review · Reviewer_iCmr · 2025-10-28

**Soundness:** 2
**Presentation:** 3
**Contribution:** 2
**Rating:** 4
**Confidence:** 4

**Summary:**

The paper proposes a novel knowledge distillation technique for large language models by integrating an adversarial training pipeline with a discriminator to verify the outputs of the teacher and student models.

**Strengths:**

1. The paper is well-written and easy to follow.
2. The knowledge distillation task is highly significant, especially at the current stage where large language models are too big and difficult to deploy on small or personalized devices.
3. Theoretical proofs provide a rigorous foundation for the validity and consistency of the proposed methods.

**Weaknesses:**

1. The concept of adversarial training has shown greater effectiveness in image domains, where data is continuous and such training can substantially enhance image quality. In contrast, for text generation tasks, where data is discrete, adversarial training tends to be less effective. This indicates that incorporating a GAN loss in addition to the KD loss does not yield significant improvements. Experimental results further demonstrate that the proposed method consistently performs on par with, or below, existing approaches.

2. In the experiments, it would be more informative to evaluate the performance when using a larger teacher model.

**Questions:**

1. Since the KD loss already aligns the student’s output distribution with that of the teacher in discrete space, the additional GAN loss essentially optimizes in the same direction. This raises the question of how the GAN loss can further contribute to performance improvement in this task.
2. Does any part of our paper explicitly address the challenge of "lacking signals for long-range consistency or higher-order dependencies across the sequence" (as you mentioned in introduction section)?

---

> ### Author Response · Authors · 2025-11-20
>
> Thanks a lot for your valuable comments!
>
> For weakness 1, we would like to clarify that our approach does not perform adversarial training in the discrete text space, which is indeed known to be challenging and unstable. Instead, the adversarial signal operates entirely in the continuous logits space, where teacher–student distributions can be aligned more effectively and where the limitations of text‑domain GANs do not apply. Moreover, our goal is not to use GANs to improve text quality—as in traditional text GANs—but to provide sequence‑level distribution alignment signals that complement token‑level KD. Finally, the relatively modest gains on smaller models are expected due to capacity limitations, whereas on longer and more structurally complex generation tasks, the benefits of this global sequence‑level signal become clear and consistent.
>
> For weakness 2, using a larger teacher model would indeed be interesting, but the computational cost of distilling multiple student sizes from substantially larger teachers is prohibitive within our resource budget. At the same time, Qwen3‑8B is already one of the strongest open‑source models in its parameter range, outperforming or matching other widely used models of similar or even larger scales. Thus, it serves as a representative and sufficiently strong teacher for evaluating distillation quality.
>
> For question 1, While the KD loss aligns the student with the teacher at the token‑level—i.e., matching conditional distributions p(y_t | x, y< t) and q(y_t | x, y< t)—it does not capture the global structure of an entire sequence. In contrast, the adversarial loss evaluates full logits trajectories, enabling the discriminator to provide sequence‑level feedback on long‑range dependencies, global coherence, and structural patterns that token‑wise KL objectives cannot model. These two signals are therefore complementary rather than redundant. This distinction is also reflected in our experimental results: on short‑answer datasets where local alignment dominates, GAKD shows limited gains over KL‑based baselines, whereas on long‑form generation tasks such as S‑NI and UnNI—where global sequence structure is essential—the adversarial component yields clear and consistent improvements.
>
> For question 2, our paper explicitly addresses the challenge of lacking long‑range or higher‑order sequence‑level signals. We clarify this in two main places: (1) in Section 3.1–3.3, where we describe how the discriminator processes entire logits sequences and provides sequence‑level adversarial gradients, which directly supply the missing global feedback beyond token‑wise KL; and (2) in our experiments, where improvements are concentrated on long‑form generation tasks (S‑NI and UnNI), demonstrating that the proposed sequence‑level signal effectively enhances long‑range consistency. That said, we agree that our exposition can be made clearer, and we will revise the manuscript to more explicitly connect this motivation with the design of the discriminator and the empirical findings.

---

### Official Review · Reviewer_UAfz · 2025-10-29

**Soundness:** 3
**Presentation:** 3
**Contribution:** 2
**Rating:** 6
**Confidence:** 3

**Summary:**

The authors propose GAN-style knowledge distillation, where they train a generator and a discriminator. The generator (student) is encouraged to generate logits that cannot be distinguished from the teacher logits with the discriminator. Results suggest the proposed method is better than existing divergence-based distillation methods.

**Strengths:**

1. The proposed method is reasonable: the authors adopt relativistic GAN, which helps them avoid mode collapse issues.
2. The authors propose importance sampling to reduce sampling cost with RKL, which is a nice touch.

**Weaknesses:**

1. The improvement seems somewhat inconsistent, specifically with smaller models. Results with different methods are generally close.
2. ROUGE-L and BLEU scores may not be the best evaluation method. Given how unstable these metrics are, I recommend the authors try some model-based metrics.
3. Ablation on different GAN variants suggest that the effect of mode collapse may not be as big as the authors claim.

**Questions:**

How efficient is the proposed method in terms of distillation, compared to other methods?

---

> ### Author Response · Authors · 2025-11-20
>
> Thanks a lot for your valuable comments!
>
> For weakness 1, the relatively modest gains on smaller student models are expected and consistent with prior findings: low‑capacity models have limited ability to absorb fine‑grained distributional signals, which naturally causes different distillation methods to converge to similar performance. As model capacity increases, these constraints diminish and the benefits of stronger distillation signals become more apparent. In our case, GAKD provides sequence‑level global feedback that is especially valuable for tasks requiring long, open-end answers. Accordingly, on benchmarks with longer answers—such as S‑NI and UnNI—GAKD yields clear and consistent improvements over all baselines, and these gains grow larger with increasing model size.
>
> For weakness 2, we appreciate the reviewer’s suggestion regarding evaluation metrics. While ROUGE‑L and BLEU indeed have limitations, they remain the standard and widely adopted metrics in prior LLM distillation work, including MiniLLM, GKD, and many recent baselines, which ensures fair and direct comparison with existing methods. More importantly, across all datasets and model sizes, GAKD consistently outperforms these strong baselines under the same evaluation conditions, indicating that the observed improvements are robust and not tied to a specific metric choice. We agree that model‑based evaluators can provide complementary perspectives, but due to their substantial computational cost—especially when evaluating multiple model sizes and benchmarks—we are unable to include them within the scope of this submission.
>
> For weakness 3, in fact, mode collapse is inherently less severe in our setting because adversarial training is not the sole optimization signal: the student is simultaneously guided by the NLL loss and the reverse KL loss, both of which act as strong anchors that constrain the model and prevent it from collapsing to degenerate modes. This makes the overall optimization fundamentally more stable than standard GAN training, and thus the differences among GAN variants in our ablations are naturally less pronounced. We acknowledge that our original writing overstated the role of mode collapse, and we will revise the paper to clarify that our choice of RGAN is motivated primarily by its smoother training dynamics and stability benefits in this multi‑objective distillation setting, rather than by mode collapse being a dominant issue.
>
> For question 1,  we list part of the runtime analysis in the Appendix (A.2). Take the student model of Qwen-4B as an example, the baseline method SFT w/o KD is trained for 5 hours on a A100 (80G) for 3 epochs; for GADK, we trained it on 4 A100 GPUs (80G) for 20 epochs for 10 hours. More detailed statistics can be found in A.2.

---

### Official Review · Reviewer_HUfL · 2025-10-31

**Soundness:** 2
**Presentation:** 3
**Contribution:** 2
**Rating:** 4
**Confidence:** 3

**Summary:**

This paper proposes a new framework to address a key limitation in existing knowledge distillation (KD) methods. Traditional techniques like Kullback-Leibler Divergence (KLD) only provide "token-wise" feedback, lacking a global, "sequence-level" signal.

GAKD solves this by adopting a Generative Adversarial (GAN) strategy. It trains the student model (the "generator") using a joint objective: 1) a reverse KLD loss for local, token-level alignment, and 2) a sequence-level adversarial loss from a "discriminator" that learns to distinguish between teacher and student outputs. This dual-feedback mechanism enables the student to better match the teacher's overall distribution, showing superior performance, especially in long-text generation tasks.

**Strengths:**

1. Novel Framework: It proposes the GAKD framework, innovatively combining token-wise KLD loss with a sequence-level adversarial (GAN) loss.

2. Superior Long-Text Performance: This dual-feedback mechanism effectively solves the problem of missing global signals, leading to better alignment and performance, especially in long-text generation.

**Weaknesses:**

1. The motivation "KLD objective only provides token-wise feedback during knowledge distillation, lacking long-range, sequence-level signals and leading to poor distribution alignment between the teacher and student models" is questionable. In MiniLLM[1] and GKD[2], the long-range information is already considered by on-policy optimization. For example, MiniLLM directly minimizes the **sequence-level** reverse KLD, and the final gradient also includes a term targeting the long-range information. Although the empirical experiments in the paper show that GAKD outperforms the baselines, more experiments are needed to justify whether the improvement is achieved by considering the long-range signals.

2. More details about the discriminator are needed. For example, how large the model is and how much additional computation cost it would introduce during KD.

**Questions:**

See Weakness.

---

> ### Author Response · Authors · 2025-11-19
>
> Thanks a lot for your valuable comments!
>
> For weakness 1, we agree that the current statement suggesting that MiniLLM and GKD do not incorporate sequence‑level information through on‑policy optimization is inaccurate, and we will revise the wording accordingly. Our intended point is not that these methods lack any sequence‑level signal, but rather that their reliance on student‑generated samples makes such signals less stable and less aligned with the teacher distribution in early training rounds—an issue our approach directly addresses. Importantly, our experiments already provide strong evidence that the performance gains of GAKD stem specifically from its sequence‑level component. Across all model sizes, GAKD shows limited or no advantage over baselines on short‑answer datasets, while consistently yielding substantial improvements on long‑text benchmarks (S‑NI and UnNI), even surpassing the teacher model. Further, ablations show that removing or weakening the adversarial sequence‑level term significantly degrades performance, and that stronger discriminators yield larger gains only on long‑text tasks. These consistent patterns across datasets, model sizes, and ablation variants demonstrate that GAKD’s improvements are indeed attributable to the incorporation of strong sequence‑level signals.
>
> For weakness 2, we introduced the details of the discriminator model in Appendix A.3, the subsection “IMPACT OF DISCRIMINATOR MODEL SIZE”. We compare two model architectures for the discriminator, namely, Qwen3-0.6B and Qwen3-1.7B. For additional computational cost, we list part of the runtime analysis in the Appendix (A.2). Take the student model of Qwen-4B as an example, the baseline method SFT w/o KD is trained for 5 hours on a A100 (80G) for 3 epochs; for GADK, we trained it on 4 A100 GPUs (80G) for 20 epochs for 10 hours. More detailed statistics can be found in A.2. Since the discriminator is not trained in the token-level but the sequence-level (the output logits of the whole sequence are used as the input to the discriminator), the training of the discriminator only accounts for a small portion of the total training runtime.

---

### Official Review · Reviewer_GjnC · 2025-11-05

**Soundness:** 2
**Presentation:** 3
**Contribution:** 3
**Rating:** 4
**Confidence:** 4

**Summary:**

This paper proposes Generative Adversarial Knowledge Distillation (GAKD), a white-box KD framework for LLMs. GAKD reframes distillation as a minimax adversarial game , training a Student (Generator) to create logits that a Discriminator cannot distinguish from the Teacher's. The Student's objective is a composite loss combining a sequence-level adversarial loss, a token-level Reverse KLD loss, and a standard Negative Log-Likelihood loss . The paper includes a proof (Corollary 1) to support optimizing the Reverse KLD using teacher-generated sequences via importance sampling. Experiments on Qwen-3 models report that GAKD outperforms baselines , especially in long-text generation.

**Strengths:**

S1. The paper is well-motivated, addressing a significant  limitation of traditional knowledge distillation: standard token-level objectives (e.g., token-wise KL) lack the long-range, sequence-level signals required for generating coherent, long-form text.

S2. GAKD demonstrates some empirical advantage over the specific baselines chosen for comparison. The method consistently outperforms off-policy token-level objectives (SFT, Supervised KD) and the MiniLLM framework on the reported instruction-following tasks (On-policy is not compared tho).

**Weaknesses:**

W1. Flawed Motivation and Missing Baselines: The paper's core motivation rests on the claim that on-policy sampling (from the student) is a "drawback" that "adversely affect[s]" performance. This assertion is outdated and mischaracterizes the current state of KD literature. A large body of recent works (e.g., GKD (https://arxiv.org/pdf/2306.13649), SKD (https://arxiv.org/abs/2410.11325)) is built specifically on on-policy sampling, proposing various methods to stabilize it (sft student model or interleaved sampling). Authors failed to compare to those approaches and justify their claims.

W2. Unjustified Adversarial Complexity: The discriminator ($D_{\phi}$) is functionally a learnable reward model that provides a sequence-level score. The paper fails to justify why this signal must be delivered via a complex, unstable adversarial minimax game. A simpler baseline would be to train the discriminator as a static reward model and add its score directly to the loss (GKD (https://arxiv.org/pdf/2306.13649) already has this study). By not comparing against this simpler alternative, the paper fails to prove that the adversarial component provides any benefit.

W3. Critically Confounded Ablation: The paper's central claim is that its adversarial sequence-level approach is superior to token-level methods. However, the GAKD loss function introduces two new signals at once: a sequence-level RKLD (via importance sampling) and a sequence-level adversarial loss. The experiments never disentangle these two effects. A critical ablation with the adversarial weight ($\beta$) set to zero is missing. We cannot know if the performance gain comes from the sequence-level RKLD term or the (unjustified) adversarial loss.

W4. Lack of Stability Analysis: The paper's own ablations (Table 2, Figure 3) show that performance is highly sensitive to the $\alpha$ and $\beta$ hyperparameters. For example, dollyeval needs low $\beta$ while SNI needs relatively higher $\beta$. The paper lacks a rigorous analysis of training stability, hyperparameter sensitivity, or potential mode collapse, all of which are well-known and critical failure modes for GANs. The method doesn't seem to generalize to unseen tasks.

W5: The sequence-level importance weight $w(x,y) = \prod_{t=1}^{T} \frac{q_{\theta}(y_t | \cdot)}{p(y_t | \cdot)}$ is a product of many ratios. For any non-trivial sequence length $T$, this product is numerically unstable and prone to vanishing (if $q_{\theta} < p$) or exploding (if $q_{\theta} > p$). The paper fails to adequately discuss how this instability is managed (e.g., log-space computation, gradient clipping) to ensure stable training.

**Questions:**

1. What is the computation overhead of this approach compared to baselines?

---

> ### Author Response · Authors · 2025-11-19
>
> Thanks a lot for your valuable comments!
>
> For W1, our goal is not to claim that on‑policy sampling methods such as GKD and SKD are obsolete, but to highlight a practical limitation in the reverse‑KL, white‑box LLM setting we study. Our statement that on‑policy sampling “adversely affects” performance is meant in this practical sense: reverse KL requires expectations under the student distribution, and for our setting , repeatedly sampling full sequences from the evolving student is computationally expensive; moreover, in early training the student often generates low‑quality sequences that the teacher rarely visits, making reverse‑KL optimization on such trajectories inefficient and potentially unstable. Recent works like GKD and SKD explicitly design on‑policy and speculative / interleaved sampling strategies to mitigate these issues. Our method is orthogonal: we show that reverse KL can be optimized using only teacher / data samples via importance sampling, while a discriminator over logits sequences supplies sequence‑level feedback for long‑range consistency. Although GKD and SKD are relevant baselines, running additional full‑scale experiments with these methods on our training corpus and model sizes would incur substantial extra computational cost beyond our hardware and time budget, so we instead focused on strong white‑box KD baselines under the same training loop.
>
> For W2, we agree that $D_{\phi}$ can be seen as a learnable sequence‑level scorer, but our goal is not just “GKD + a static reward model”. The key is that we train $D_{\phi}$ adversarially and continuously against the current student, rather than fixing it after a one‑time training. A static reward model quickly becomes miscalibrated under distribution shift: as the student changes, the scorer is no longer optimized to distinguish current student outputs from teacher‑like ones, so it can provide outdated or misleading gradients, especially when combined with reverse‑KL. In contrast, our adversarial formulation keeps $D_{\phi}$ updated on the trajectories the student actually produces, yielding a sequence‑level signal that stays aligned with the current teacher–student mismatch and emphasizes global structural errors in long texts. Empirically, under the same data and compute, GAKD already outperforms strong non‑adversarial baselines.
>
> For W3, we clarify that RKLD loss (via importance sampling) fundamentally remains a token-level objective, as it optimizes the divergence of conditional probabilities at each time step rather than evaluating the complete sequence holistically. Therefore, it cannot replace the sequence-level semantic feedback provided by the discriminator. This distinction is empirically supported by Table 2: as $\beta$ decreases from 0.1 to 0.01, performance consistently declines on long-text datasets (e.g., Rouge-L of UnNI: 42.918 to 42.685). This trend underscores the importance of the global adversarial signal for maintaining generation quality, confirming that the discriminator is essential.
>
> For W4, we clarify that the varying optimal $\beta$ values reflect an interpretable trade-off between token-level and sequence-level guidance tailored to specific task characteristics rather than model instability. Specifically, since DollyEval consists largely of short, closed-ended responses, a lower $\beta$ is preferred to allow token-level feedback (NLL) to dominate for precision. Conversely, SNI involves longer, open-ended generation where a higher $\beta$ is necessary to strengthen sequence-level feedback from the discriminator, which is crucial for maintaining global coherence in long sequences. Furthermore, unlike vanilla GANs, our inclusion of the sequence-level RKLD (via importance sampling) acts as a regularization anchor—similar to the KL penalty in PPO—which constrains the policy to the reference distribution and effectively prevents mode collapse.
>
> For W5, in our actual implementation (see provided code), we do not compute the raw product of importance weights $\prod (q/p)$. Instead, we utilize a Reverse KL-Divergence objective computed in log-space (sum(q * (log q - log p))). This transforms the potentially unstable product into a stable summation, fundamentally avoiding arithmetic overflow/underflow issues.
>
> For Q1, we list part of the runtime analysis in the Appendix (A.2). Take the student model of Qwen-4B as an example, the baseline method SFT w/o KD is trained for 5 hours on a A100 (80G) for 3 epochs; for GADK, we trained it on 4 A100 GPUs (80G) for 20 epochs for 10 hours. More detailed statistics can be found in A.2.

---

### Meta-Review · Area_Chair_DwDQ · 2025-12-17

**Summary:**

Three reviewers (Reviewer GjnC, Reviewer HUfL and Reviewer iCmr) claimed that the authors fail to validate their motivation about the drawback of on-policy sampling due to missing comparative experiments. Reviewer UAfz agreed to the rationality of the proposed method. Reviewer iCmr and Reviewer UAfz concerned about the performance gain over previous methods and the evaluation metrics, including ROUGE-L and BLEU scores.

The AC also concerned that the evaluation metrics, including ROUGE-L and BLEU scores, are not very popular benchmarks to validate the effectiveness of LLMs. Moreover, further comparative experiments with on-policy sampling can also refine the soundness of this paper.

**Reviewer Concerns:**

The concerns about the effectiveness validation against on-policy sampling and evaluation metrics are not well addressed.

**Reviewer Scores:**

Reviewer GjnC: 4;

Reviewer HUfL: 4;

Reviewer UAfz: 6;

Reviewer iCmr: 4

---

### Decision · Program_Chairs · 2026-01-26

Reject